# Suppression of PGC-1α Drives Metabolic Dysfunction in TGFβ2-Induced EMT of Retinal Pigment Epithelial Cells

**DOI:** 10.3390/ijms22094701

**Published:** 2021-04-29

**Authors:** Daisy Y. Shu, Erik R. Butcher, Magali Saint-Geniez

**Affiliations:** 1Schepens Eye Research Institute of Mass, Eye and Ear, Department of Ophthalmology, Harvard Medical School, Boston, MA 02114, USA; daisy_shu@meei.harvard.edu; 2Harvard John A. Paulson School of Engineering and Applied Sciences, Harvard University, Boston, MA 02134, USA; erik_butcher@meei.harvard.edu

**Keywords:** retinal pigment epithelium (RPE), metabolism, mitochondria, transforming growth factor-beta (TGFβ), epithelial-mesenchymal transition (EMT), bioenergetics, PGC-1α, OXPHOS, glycolysis, mitochondrial dynamics

## Abstract

PGC-1α, a key orchestrator of mitochondrial metabolism, plays a crucial role in governing the energetically demanding needs of retinal pigment epithelial cells (RPE). We previously showed that silencing *PGC-1α* induced RPE to undergo an epithelial-mesenchymal-transition (EMT). Here, we show that induction of EMT in RPE using transforming growth factor-beta 2 (TGFβ2) suppressed *PGC-1α* expression. Correspondingly, TGFβ2 induced defects in mitochondrial network integrity with increased sphericity and fragmentation. TGFβ2 reduced expression of genes regulating mitochondrial dynamics, reduced citrate synthase activity and intracellular ATP content. High-resolution respirometry showed that TGFβ2 reduced mitochondrial OXPHOS levels consistent with reduced expression of *NDUFB5*. The reduced mitochondrial respiration was associated with a compensatory increase in glycolytic reserve, glucose uptake and gene expression of glycolytic enzymes (*PFKFB3*, *PKM2*, *LDHA*). Treatment with ZLN005, a selective small molecule activator of PGC-1α, blocked TGFβ2-induced upregulation of mesenchymal genes (*αSMA*, *Snai1*, *CTGF*, *COL1A1*) and TGFβ2-induced migration using the scratch wound assay. Our data show that EMT is accompanied by mitochondrial dysfunction and a metabolic shift towards reduced OXPHOS and increased glycolysis that may be driven by PGC-1α suppression. ZLN005 effectively blocks EMT in RPE and thus serves as a novel therapeutic avenue for treatment of subretinal fibrosis.

## 1. Introduction

Emerging evidence implicates a role for mitochondrial and metabolic dysfunction in cancer metastasis [1,2] and tissue fibrosis [3,4], both driven by the unifying mechanism of epithelial-mesenchymal transition (EMT). During EMT, the once stationary and cuboidal epithelial cells transdifferentiate into spindle-shaped, migratory mesenchymal cells that secrete copious amounts of extracellular matrix (ECM) proteins [5]. In the eye, EMT of the retinal pigment epithelium (RPE) is a key mechanism underpinning retinal fibrotic diseases such as proliferative vitreoretinopathy (PVR) and subretinal fibrosis in age-related macular degeneration (AMD) [6]. Transforming growth factor-beta 2 (TGFβ2) is a potent inducer of EMT in RPE. Elevated TGFβ2 levels have been detected in the vitreous of patients with PVR [7,8,9] and contributes to increased collagen synthesis and deposition in PVR eyes [10]. All three mammalian TGFβ isoforms and TGFβ receptor type II have been identified in epiretinal membranes associated with PVR [11].

However, to date, there is no literature on whether or how mitochondrial morphology, function or bioenergetic profiles are altered during the induction of EMT in RPE. Work in our laboratory showed that repression of *PGC1α*, a master regulator of mitochondrial biogenesis and metabolic function, in human RPE cells disrupted mitochondrial function, redox state, energy sensor activity and autophagy function, as expected, but surprisingly also induced an EMT response [12]. Given that induction of mitochondrial dysfunction induced a cell-fate switch, driving cells into an EMT, we asked the inverse question: does EMT activation in RPE cells result in any mitochondrial or metabolic defects?

While the precise relationship between metabolic plasticity and EMT remains elusive, it is evident that the morphological and biomolecular changes during EMT coupled with the enhanced cellular motility of mesenchymal cells must require different cellular bioenergetics and metabolic coordination [13]. Cancer cells are well-known to exhibit metabolic plasticity and favor aerobic glycolysis known as the “Warburg effect” [14,15]. This metabolic switch confers a survival advantage for cancer cells to exploit the surrounding nutrients, rapidly generate ATP, synthesize biomass and balance reactive oxygen species (ROS) [14]. In the present study, we treated ARPE-19 and primary human RPE (H-RPE) cells with TGFβ2, a key inducer of EMT and conducted a series of metabolic assays including analysis of mitochondrial network morphology, mitochondrial dynamics and real-time bioenergetic profiling.

## 2. Results

### 2.1. TGFβ2-Induced EMT of ARPE-19 Is Accompanied by Mitochondrial Dysfunction

Untreated control ARPE-19 retained a regular polygonal morphology while TGFβ2 induced ARPE-19 to elongate into spindle-shaped mesenchymal cells at 48 h (Figure 1A). Confirmation of EMT was validated using immunofluorescence and qPCR for EMT markers. Control ARPE-19 showed minimal protein expression of vimentin whereas TGFβ2 induced an upregulation of vimentin expression that appeared condensed in prominent perinuclear intermediate filament bundles (Figure 1A). TGFβ2 significantly upregulated expression of EMT genes involved in ECM remodeling including collagen 1A1 (*COL1A1*; Figure 1C), fibronectin (*FN1*; Figure 1D), connective tissue growth factor (*CTGF*; Figure 1E) and matrix metalloproteinase-2 (*MMP2*; Figure 1F) over 72 h.

Silencing *PGC-1α* induced ARPE-19 to undergo EMT [12] in the absence of any exogenous stimulus suggesting that suppression of this metabolic transcriptional co-factor is critical in promoting EMT. Thus, we investigated the effect of TGFβ2-mediated EMT on the expression of *PGC-1α*. We found that TGFβ2 significantly suppressed *PGC-1α* over the 72-h treatment period (Figure 1G). Our previous study showed that silencing *PGC-1α* in ARPE-19 also induced disorganization of the mitochondrial network with loss of tubular structure and acquisition of donut/blob morphology [12], identified as hallmarks of mitochondrial dysfunction [16]. Here, we showed that TGFβ2 also induced a disruption of mitochondrial network morphology. Control cells displayed an elongated and filamentous mitochondrial network, whereas TGFβ2-treated cells exhibited a smaller, fragmented and spherical mitochondrial network (Figure 1B). We adapted the computational methodology described by Nikolaisen et al. (2014) [17] for unbiased and automated 2D and 3D extraction and quantification of mitochondrial shape and network morphology (Figure 1H). Consistent with the observations of captured z-stack images of the mitochondrial network, TGFβ2 significantly reduced the total volume, mean volume and the surface area of mitochondria compared to untreated control cells (Figure 1H). An increase in sphericity and reduction in the number of branches, total branch length and mean branch length was also observed following TGFβ2 treatment indicating a disruption of the normally elongated and filamentous mitochondrial network. No significant changes were observed in branch end points or mean branch diameter.

Mitochondrial morphology depends upon the state of mitochondrial dynamics, a coordinated balance of mitochondrial fusion and fission [18]. TGFβ2 treatment disturbed this finely tuned balance by increasing mitochondrial fusion, upregulating both *MFN1* and *MFN2* (Figure 1I) and the mitochondrial fission gene, *FIS1* at 48 h (Figure 1J).

### 2.2. TGFβ2 Reduces Mitochondrial Respiration in RPE

Since mitochondria are the site of OXPHOS, we next investigated whether TGFβ2 altered the bioenergetic profile of ARPE-19 using the Seahorse XF24 BioAnalyzer. Real-time oxygen consumption rate (OCR) was measured following sequential injections of electron transport chain (ETC) inhibitors and mitochondrial respiration parameters were calculated. TGFβ2 significantly reduced the spare respiratory capacity and maximal respiration levels at both 24 h (Figure 2A,B) and 72 h (Figure 2C,D). There were no changes in basal respiration, ATP-linked respiration, or proton leak.

During OXPHOS, electrons from oxidative substrates generated in the tricarboxylic acid (TCA) cycle are transferred to oxygen through a series of redox reactions to form water. In this process, protons are pumped from the matrix across the mitochondrial inner membrane through respiratory complexes of the ETC that produce an electrochemical gradient to enable ATP generation. TGFβ2 significantly downregulated gene expression of *NDUFB5* (a key component of Complex I) at 24 h (Figure 2E).

Since TGFβ2 disrupted normal mitochondrial morphology and dynamics, we next asked whether it also affected mtDNA levels. Mitochondria are highly dynamic organelles whose biogenesis is under tight nuclear regulation, thereby conferring mitochondrial adaptability to changes in cellular milieu [1]. TGFβ2 had no effect on mitochondrial copy number at 24 or 72 h (Figure 2F).

Citrate synthase is the first enzyme of the TCA cycle, catalyzing the conversion of acetyl-CoA into citrate in the mitochondria. Changes in the activity of citrate synthase can serve as a biomarker of changes in mitochondrial mass [19]. Consistent with our observations of mitochondrial morphology and function, TGFβ2 slightly suppressed citrate synthase activity at 24 h (Figure 2G); however, this effect was not maintained at 48 or 72 h. A substantial amount of energy is generated in mitochondria through OXPHOS (~36 mol ATP compared to 2 mol ATP in glycolysis per mol of glucose). Consistent with the reduced mitochondrial respiration, a significant reduction in intracellular ATP content was observed at 72 h (but not 24 or 48 h) following TGFβ2 treatment (Figure 2H).

### 2.3. TGFβ2 Enhances Glycolytic Reserve in RPE and Increases Glycolysis Gene Expression

Given the reduction in mitochondrial respiration, we next asked whether there were any compensatory bioenergetic shifts in the glycolytic pathway. Real-time analysis of glycolytic function using the Seahorse XF24 BioAnalyzer measures changes in extracellular acidification rate (ECAR). While no significant differences in basal glycolysis were observed, TGFβ2 significantly upregulated glycolytic capacity and glycolytic reserve at 24 h (Figure 3A,B) and 72 h (Figure 3C,D). TGFβ2 increased glucose uptake (Figure 3E) and increased gene expression of transporters and enzymes critical to the glycolytic pathway. Specifically, TGFβ2 significantly upregulated gene expression of the glucose transporter *GLUT3*, but not *GLUT1* or *GLUT12* (Figure 3F). Cancer cells that show a preference for glycolysis have been observed to upregulate the expression of monocarboxylate transporters (MCTs) to serve as lactate shuttles [20]. The RPE is known to express two MCTs, MCT1 apically and MCT3 basolaterally [21]. We found that TGFβ2 significantly upregulated the gene expression of *MCT1* but not *MCT3* (Figure 3G). Additionally, we found that TGFβ2 also significantly increased the gene expression of three key glycolytic enzymes, specifically *PFKFB3* (Figure 3H), *PKM2* (Figure 3I) and *LDHA* (Figure 3J).

### 2.4. TGFβ2 Suppresses PGC-1α Gene Expression and ZLN005 Blocks TGFβ2-Induced EMT in ARPE-19 and Primary Human RPE

We previously showed that ZLN005, the selective small molecular activator of PGC-1α, effectively increased *PGC-1α* gene expression in ARPE-19 and enhanced mitochondrial respiration [22]. Accordingly, we determined whether *PGC-1α* activation by ZLN005 could block TGFβ2-induced EMT in RPE. Phase-contrast microscopy showed that ZLN005 suppressed the spindle morphology and vimentin upregulation induced by TGFβ2 (Figure 4A,B). ZLN005 robustly enhanced *PGC-1α* gene expression in control ARPE-19 cells (6-fold increase; Figure 4C) and maintained this elevation even in the presence of TGFβ2 (5.6-fold increase). *PGC-1α* augmentation by ZLN005 blocked TGFβ2-induced upregulation of mesenchymal gene expression, specifically *α-SMA* (Figure 4D), *Snai1* (Figure 4E), *CTGF* (Figure 4F) and *COL1A1* (Figure 4G). ZLN005 also significantly suppressed TGFβ2-induced migratory activity as imaged by phase-contrast microscopy (Figure 4H), quantified as the percentage of wound closure (Figure 4I). Importantly, the inhibitory effect of ZLN005 on migration was observed in non-TGFβ2-activated cells indicating that the effect of ZLN005 is not only TGFβ2-dependent, but that ZLN005 also blocks EMT activation triggered by scratch-wounding in control cells.

Similar findings were observed in primary human RPE cells (H-RPE). TGFβ2 induced an upregulation of mesenchymal gene expression (*COL1A1*, *FN1*, *MMP2*) at 24 h (Figure 5A) and this was accompanied by an upregulation in glycolytic gene expression (*PFKFB3*, *LDHA*, *PKM2*; Figure 5B). TGFβ2 also suppressed *PGC-1α* gene expression in H-RPE at 24 h (Figure 5C). Real-time analysis of glycolytic function using the Seahorse XFe96 BioAnalyzer of H-RPE showed that TGFβ2 treatment for 5 days significantly upregulated glycolytic capacity (Figure 5D,E). Administration of 10 μM ZLN005 blocked TGFβ2-induced upregulation of mesenchymal protein expression of vimentin and α-SMA (Figure 5F). ZLN005 also increased *PGC-1α* gene expression in control and TGFβ2-treated H-RPE at 24 h (1.3-fold increase, Figure 5G), albeit not as robustly as the increase observed in ARPE-19. However, on comparing the basal levels of *PGC-1α* gene expression between ARPE-19 and H-RPE, it is evident that *PGC-1α* levels are inherently greater in H-RPE (8-fold higher, Figure 5H) and may help to explain the differences in ZLN005 upregulation of *PGC-1α* expression between these cells.

## 3. Discussion

We report that TGFβ2-induced EMT in RPE is accompanied by a disruption of the mitochondrial network integrity and a rewiring of the bioenergetic profile towards glycolysis. TGFβ2 depleted *PGC-1α* gene expression, a critical mediator of mitochondrial biogenesis and metabolic function in RPE [23], building on our previous work showing that silencing *PGC-1α* also induced mitochondrial dysfunction and subsequent EMT [12]. We show that selective induction of PGC-1α using ZLN005 effectively blocks TGFβ2-induced EMT markers and migratory function, indicating the importance of PGC-1α in driving EMT of RPE.

The disruption of mitochondrial network integrity following TGFβ2 treatment in RPE was associated with an imbalance of mitochondrial dynamics genes. Specifically, we observed an increased expression of mitochondrial fusion genes, *MFN1* and *MFN2* and the mitochondrial fission gene, *FIS1*, with TGFβ2. Mitochondrial hyperfusion can be triggered upon stress by sharing neighboring mitochondrial ETC components to compensate for defective protein complexes required for ATP production, thereby restoring mitochondrial integrity and metabolic efficiency [24,25]. Indeed, TGFβ2 reduced both metabolic function (reduced maximal respiration and spare respiratory capacity) and expression of *NDUFB5*, a critical component of Complex I of the ETC known to play a role in maintaining the mitochondrial network architecture [26]. Taken together with the reduced *PGC-1α* levels, it is possible that the defective mitochondria may be stimulating mitochondrial fusion genes to compensate for their diminished metabolic function.

In addition to increases in mitochondrial fusion observed with TGFβ2, a concomitant increase in the mitochondrial fission gene, *FIS1*, was observed following TGFβ2 treatment. Increased fission has also been observed in a model of hepatocyte EMT liver fibrosis model and has been linked to diminished PGC-1α expression [4]. The dysregulation of both mitochondrial fission and fusion genes with TGFβ2 treatment in RPE correlates morphologically with the fragmented mitochondrial network architecture exhibiting reduced branching and increased sphericity, hallmarks of mitochondrial dysfunction [17,27].

Organization of the mitochondrial network is a strong indicator of bioenergetic capacity [28,29] and cellular health [30,31]. The disrupted mitochondrial network morphology with TGFβ2 treatment was accompanied by a reduction in spare respiratory capacity, a measure of the respiratory potential of a cell to balance the sudden fluctuations in energy demands during oxidative stress [32] or defects in ETC complexes [33]. Notably, there were no changes in mitochondrial DNA copy number with TGFβ2 indicating that diminished mitochondrial respiration potential was due to a dysfunction of the mitochondria, rather than the mitochondrial genome. Mitochondrial OXPHOS produces ATP with far greater efficiency compared to glycolysis. In our study, we showed that TGFβ2 also induced a severe depletion of cellular ATP content at 72 h indicating that the defective mitochondria were directly affecting important cellular functions such as ATP production. We also observed a reduction in the activity of citrate synthase, the first enzyme of the TCA cycle, following TGFβ2 treatment further supporting the generalized mitochondrial and metabolic dysfunction induced by TGFβ2.

Intriguingly, we observed an increase in glycolytic capacity and reserve following TGFβ2 treatment which was accompanied by an increased expression of glycolytic enzymes, *PFKFB3*, *PKM2* and *LDHA*, linked to EMT in cancer metastasis models [34,35]. PFKFB3, a potent activator of the rate limiting enzyme in glycolysis, PFK-1, that catalyzes the irreversible conversion of fructose-6-phosphate into fructose-1,6-bisphosphate is reported to be an effector protein mediating TGFβ1-induced EMT in tumor cells [36] and its expression increases during tumorigenesis [37]. PKM2, another rate-limiting glycolytic enzyme that catalyzes the irreversible conversion of phosphoenolpyruvate (PEP) and ADP into pyruvate and ATP, is overexpressed in cancer and plays a key role in aerobic glycolysis [38].

Cancer studies have reported extensively on the “Warburg effect” whereby cells prefer aerobic glycolysis despite intact and fully functional mitochondrial machinery [39]. Although ATP production from glycolysis may appear to be a highly inefficient form of energy production, tumors can benefit from this metabolic switch. Firstly, glycolysis does not consume oxygen and thus, tumors can increase their tolerance to fluctuations in oxygen levels. Secondly, lactate (the end product of glycolysis), can increase the acidity of the cellular milieu, thereby triggering the activation of various growth factors including TGFβ2 [40,41] and vascular endothelial growth factor (VEGF) [42] that enhance tumor invasiveness, angiogenesis and perpetuate the vicious cycle. The enhanced lactate can also further stimulate TGFβ-induced migratory activity [43]. Finally, aerobic glycolysis can be utilized to provide substrates for biosynthetic pathways to balance the demands of rapidly proliferating cancer cells [39].

The metabolic shift that occurs in RPE following TGFβ2 treatment lends us to consider how this will impact the neighboring retinal and choroidal cells and ultimately, the metabolic ecosystem of the eye. In the retina, enhancing glycolysis appears to be beneficial for photoreceptors by enhancing their robustness and delaying degeneration [44,45]. In contrast, increasing glycolysis in RPE has a detrimental effect, causing the neighboring photoreceptors to degenerate [46,47,48]. Therefore, TGFβ2-induced enhanced glycolysis in RPE during EMT may trigger a detrimental secondary effect to photoreceptors. The main metabolic pathway utilized by RPE cells is reductive carboxylation, important for supporting redox homeostasis [49] and how this pathway is impacted by TGFβ2 will be explored in future metabolomics analysis.

We found an elevated expression of the glucose transporter, *GLUT3* and the lactate transporter, *MCT1*, with TGFβ2 in RPE. Increased glucose transporter expression is a key driver of increased glycolysis in malignant cancer [50] with GLUT3 contributing directly to enhanced glucose uptake and cancer progression through EMT induction [51]. Elevated MCT-1 expression supports the glycolytic preference of cells by enabling the efficient export of lactate, thereby minimizing potential cellular stress associated with excessive acid accumulation [52]. The elevated *MCT1* levels following TGFβ2 treatment in RPE in our study may have detrimental effects on the retinal microenvironment, as MCT-1 is a critical regulator of water homeostasis in RPE and plays a role in maintaining intracellular pH of RPE [53,54]. In the eye, knockout of MCT3 resulted in a build-up of lactate in RPE and the inter-photoreceptor matrix, disturbing the ionic homeostasis of the outer retina and impairing visual function [55].

ZLN005 exerts numerous protective effects on RPE including increased expression of antioxidant enzymes, decreased mitochondrial superoxide production and enhanced OXPHOS [22]. The inhibitory effect of ZLN005 on TGFβ2-induced EMT in our study reveals that promoting *PGC-1α* induction in RPE is sufficient to block EMT. It is likely that PGC-1α does not directly repress EMT genes and thus, we speculate that PGC-1α may be working indirectly to inhibit EMT by activating mitochondrial biogenesis, increasing antioxidant activity or operating synergistically with sirtuins, a family of proteins known to protect against metabolic stress [56]. Sirtuin1 (SIRT1) is a well-established activator of PGC-1α [57]. Ishikawa et al. (2015) showed that promoting SIRT1 activity using resveratrol effectively blocked TGFβ2-induced EMT of RPE in a rabbit PVR model [58]. Indeed, evidence in cardiomyocytes showed that ZLN005 exerts protective effects against high glucose by promoting SIRT1 expression [59]. In models of breast cancer metastases and kidney fibrosis, SIRT1 suppressed EMT by deacetylating Smad4 and thus, repressing TGFβ-induced MMP7 [60]. Moreover, pharmacological activation of SIRT1 using the small molecule SRT1720 attenuated cardiac fibrosis by targeting TGFβ-induced Smad2/3 transactivation [61]. Future investigations are required to identify the precise mechanistic insights underpinning the inhibitory effect of ZLN005 on EMT of RPE.

Taken together, our data show that TGFβ2-induced EMT in RPE is associated with mitochondrial dysfunction and represents a new paradigm for understanding retinal fibrosis. Metabolic reprogramming is emerging as a key driver of EMT in cancer metastasis and now we have extended these findings to the pathogenesis of PVR and AMD. We highlight a role for PGC-1α, a master regulator of mitochondrial biogenesis and energy expenditure, as a crucial metabolic node underpinning this metabolic rewiring process. What unfolds here is a novel therapeutic avenue: EMT could be potentially blocked by promoting mitochondrial health and metabolic function and/or inhibiting metabolic rewiring towards glycolysis. We show promising data for the efficacy of ZLN005 as a potential novel treatment for PVR and subretinal fibrosis in AMD. Our future studies endeavor to further characterize the distinct metabolic hallmarks exclusive to EMT of RPE to unravel critical mechanistic insights and harness the full potential of mitochondrial- and metabolic-targeting therapies for the retina.

## 4. Materials and Methods

### 4.1. Cell Culture

The human retinal pigment epithelial cell line, ARPE-19 (ATCC, Manassas, VA, USA) was expanded in growth medium comprising of DMEM/F12 (Cat. no. 11330-032; Thermo Fisher Scientific, Wilmington, DE, USA) supplemented with 10% fetal bovine serum (FBS, Atlanta Biologicals, Lawrenceville, GA) and 1% penicillin and streptomycin (PenStrep, Lonza, Walkersville, MD, USA) at 37 °C and 10% CO_2_ in a humidified incubator. Cells were detached using trypsin-EDTA. Cells were matured before seeding to confluency in multi-well plates. Cells were maintained in serum free media for 2–3 days before treatment with recombinant human TGFβ2 (Peprotech, Rocky Hill, NJ, USA) at 10 ng/mL and/or 10 μM ZLN005 (Cayman Chemicals). Maximal passage of 5 and cells were split 1:3 for each passage.

Primary human fetal retinal pigment epithelial cells were purchased from Lonza (H-RPE, Cat #00194987) and matured in RtEGM Retinal Pigment Epithelial Cell Growth Medium supplemented with RtEGM SingleQuots (4 mM L-glutamine, 25 ng/mL FGF-2, 2% FBS, 30 mg/mL gentamicin and 15 μg/mL amphotericin). Media was changed every 3–4 days and cells were detached using the ReagentPack Subculture Reagents (Lonza) consisting of Trypsin/EDTA, trypsin neutralizing solution and HEPES buffered saline solution. Cells were seeded to confluence in multi-well plates and serum starved for 2–3 days before adding 10 ng/mL TGFβ2 and/or 10 μM ZLN005.

### 4.2. Immunofluorescence Confocal Microscopy

RPE cells were seeded on glass coverslips (12 mm diameter) to confluence in a 24-well plate. Cells were serum starved for 2 days before adding TGFβ2 for 72 h. Cells were rinsed in PBS ×2 before fixation in 4% paraformaldehyde for 10 min and then rinsed in PBS (3 × 5 min). Cells were permeabilized using 0.01% Triton X-100 in PBS for 5 min and then rinsed in PBS (3 × 5 min). Cells were blocked for 1 h at RT in 1% bovine serum albumin and 3% normal goat serum diluted in PBS. Cells were incubated in the primary antibody overnight at 4 °C (rabbit monoclonal anti-vimentin at 1:200 from Cell Signaling Technology #5741S and mouse monoclonal anti-alpha smooth muscle actin at 1:400 from Sigma #A2547). The next day, cells were washed in PBS (3 × 10 min) and incubated for 2 h in the dark at RT in the secondary antibody: goat anti-rabbit Alexa Fluor 488 (1:1000; Invitrogen #A11034) and/or goat anti-mouse Alex Fluor 594 (1:1000; Invitrogen #A11032) and DAPI (1:100). Cells were rinsed in PBS (3 × 10 min). Coverslips were mounted on slides using mounting medium (ibidi #50011) and sealed with CoverGrip Coverslip Sealant (Biotium #23005). Z-stack images were taken on the Leica SP8 confocal microscope and processed using FIJI and Adobe Photoshop.

### 4.3. Quantitative PCR (qPCR)

Total RNA was extracted using E.Z.N.A.Total RNA Kit I (Omega Bio-Tech, Norcross, GA, USA) and RNA concentrations were measured using the NanoDrop Spectrophotometer ND-1000 (ThermoFisher Scientific). Only samples with 260/280 ratios > 2 were included in further analysis. To remove DNA contaminants, 1 μg of the purified RNA was treated with ezDNAse enzyme (Thermo Fisher) before being reverse transcribed into cDNA using the SuperScript IV VILO MasterMix (ThermoFisher Scientific). cDNA was diluted 1:10 and amplified by real-time PCR using the PowerUp SYBR Green Master Mix (ThermoFisher Scientific, San Jose, CA, USA) in a LightCycler 480, 384-well plate (Roche) consisting of 2× SYBR, 10 ng cDNA, 1 μM forward and reverse primers in nuclease free H_2_O (Table 1). All reactions were run in duplicate, including minus RT and no-template controls under the following thermal cycling conditions: 50 °C, 2 min; 95 °C, 2 min, followed by 40 cycles of 95 °C for 15 s and 60 °C for 1 min. Melt curve analysis was performed to confirm amplification specificity. Ct values were normalized to the housekeeping genes *PPIH*, *B2M* or *TBP* using the second derivative maximum method.

### 4.4. Mitochondrial Morphology Imaging

ARPE-19 cells were seeded in 24-well plates to 100% confluence on circle coverslips (12 mm diameter) and serum starved for 2 days before treatment with TGFβ2 for 72 h. MitoTracker Orange CMTMRos (Invitrogen, Carlsbad, CA, USA) was reconstituted to 1 mM stock in DMSO and diluted 1:10,000 in serum-free media before addition to cells for 30 min in the incubator. Cells were fixed in ice-cold methanol at −20 °C (15 min) then rinsed in PBS × 3 (5 min) before mounting on slides. Z-stack images were acquired using the Leica SP8 confocal microscope system.

### 4.5. Automated Processing of Mitochondrial Network Morphology

MitoTracker colorimetric optical quantification of the fluorescent protein-based biomarkers was completed using a custom ImageJ plugin (1.52q, FIJI, http:/imagej.nih.gov/ij, accessed on 18 December 2020, National Institutes of Health, Bethesda, MD, USA). The images were ratiometrically pre-processed using several java functions either installed natively with FIJI or as part of the custom plug-in setup process, including “Sigma Filter”, “Subtract Background”, “Enhance Local Contrast” and “Gamma Correction” to remove noise from biostructure signal. The pre-processed images were then reviewed using a proofing-sheet to optimize the local “Adaptive Threshold” algorithm block size and C-value parameters for processing. Using the batch-processing functionality, all images were thresholded and binarized to isolate the mitochondrial network biostructure signal using the determined configuration values. Once the mitochondrial network biostructures were isolated, the images were post-processed using “Remove Outliers”, “Despeckle” and “Fill 3D Holes” to remove any remaining optical artifacts. Individual mitochondrial networks were then extracted and persisted to computer memory as regions of interest (ROIs). The three-dimensional Z-stack images were analyzed using the functions from the MorphoLibJ package “3D Object Counter” and “3D Particle Analyzer”, generating per-cell or per-mitochondria (depending on run configurations) 3D quantifications of mitochondrial dynamics and morphology parameters. The raw mitochondrial morphometric data were completed in GraphPad Prism for final statistical analysis.

### 4.6. High-Resolution Respirometry

The Seahorse XF24 and XFe96 Analyzers (Agilent Technologies, Santa Clara, CA, USA) were used to determine the oxygen consumption rate (OCR) and extracellular acidification rate (ECAR) for ARPE-19 (seeded at 50,000 cells per well in V7-PS TC-Treated XF24 Cell Culture Microplates) and H-RPE (seeded at 21,000 cells per well in XFe96 Cell Culture Microplates). For the Mito Stress Test, the medium was replaced with the assay medium (Seahorse XF Base Medium without Phenol Red, Agilent) supplemented with 2 mM glutamine (Lonza), 1 mM pyruvate (Gibco, Carlsbad, CA, USA) and 25 mM glucose (Sigma, St. Louis, MO, USA), pH 7.4 and placed in a 37 °C, CO_2_-free, humidified incubator for 1 h. The drug injections were oligomycin (2.5 μM), FCCP (500 nM) or BAM15 (10 μM) and a combination of rotenone and antimycin A (both at 2 μM). For the Glycolytic Stress Test, the medium was replaced with the assay medium (Seahorse XF Base Medium without Phenol Red, Agilent) supplemented with 1 mM L-Glutamine, pH 7.4 and placed in a 37 °C, CO_2_-free, humidified incubator for 1 h. The drug injections were glucose (10 mM), oligomycin (2 μM) and 2DG (50 mM). Cells were lysed in cold 1x Cell Lysis Buffer (Cell Signaling Technology) supplemented with 1 mM PMSF (Sigma) and stored at −80 °C. Protein concentration was quantified using the Pierce BCA Assay kit (Thermo Fisher). Data were normalized to protein content using the XF Wave software by exporting the XF Mito Stress Test and XF Glycolytic Stress Test Report Generators to Excel and GraphPad Prism.

### 4.7. Quantification of Intracellular ATP Content

ARPE-19 cells were seeded in 6-well plates to confluence and serum starved for 2 days before adding TGFβ2 for up to 3 days. ATP content was quantified using the ATP bioluminescence assay kit CLS II (Roche, Heidelberg, Germany), based on the light-emitting oxidation of luciferin by luciferase in the presence of low ATP levels. Cells were rinsed in PBS × 2 (5 min) before adding 100 mM Tris buffer containing 4 mM EDTA. Cells were collected by scraping and boiled for 2 min. Samples were then centrifuged at 1000× *g* and the supernatant was removed and placed on ice. Protein content was measured using the Pierce BCA Assay kit (Thermo Fisher). A total of 150 μg of protein for each sample was run in duplicate in a white, flat bottom 96-well microplate. Determination of free ATP was performed according to the manufacturer’s protocol and compared against an ATP standard curve. Luminescence was measured at 25 ms integration using a Molecular Devices Spectramax M3 plate reader.

### 4.8. Citrate Synthase Assay

ARPE-19 cells were seeded in 6-well plates to confluence and serum starved for 2 days before adding TGFβ2 for up to 3 days. Cells were lysed in cold 1× Cell Lysis Buffer (Cell Signaling Technology) supplemented with 1 mM PMSF (Sigma). Protein concentration was quantified using the Pierce BCA Assay kit (Thermo Fisher). Citrate synthase activity was measured using the MitoCheck citrate synthase activity assay kit (Cayman Chemical, Ann Arbor, MI, USA) as per the manufacturer’s directions in a clear 96-well microplate. Absorbance was measured at 412 nm using a BioTek-Synergy 2 (BioTek Instruments, Inc., Winooski, VT, USA).

### 4.9. Mitochondrial Copy Number

ARPE-19 cells were seeded in 6-well plates to confluence and serum starved for 2 days before adding TGFβ2 for up to 3 days. Genomic DNA (gDNA) was extracted using the NucleoSpin DNA RapidLyse kit (Macherey Nagel, Düren, Germany) and DNA concentration and purity was measured using the NanoDrop Spectrophotometer ND-1000 (ThermoFisher Scientific). Purity of the extracted DNA was based on the OD 260/280 nm absorbance ratio and were between 1.8 and 2.2. Each reaction was run in duplicate and consisted of 10 ng of gDNA, 5 μL of PowerUp SYBR Green Master Mix (ThermoFisher Scientific), 1 μL of the primer mix, all diluted in water to a final volume of 10 μL. The two primers specific for determination of nuclear DNA were *SLCO2B1* and *SERPINA1* and the two primers for detection of mtDNA were *ND1* -NADH dehydrogenase subunit 1- and *ND5*, all provided in the Clontech Human Mitochondrial DNA (mtDNA) Monitoring Primer Set (Takara Bio Inc, Kusatsu, Shiga, Japan). Reactions were run on a 384-well plate using a LightCycler 480 (Roche). qPCR was performed as per the thermal cycles mentioned above. Based on the Ct values obtained for each of the four target genes, the copy number of mtDNA was defined as the average of the ratio of mtDNA to nuclear DNA using the two pairs of genes mt-*ND5* with *SERPINA1* and *ND1* with *SLCO2B1*.

### 4.10. Glucose Uptake Assay

ARPE-19 cells were seeded in white 96-well plates (Corning #3903) to confluence and serum starved for 2 days before adding TGFβ2 (10 ng/mL) for 24 h. Glucose uptake was measured using the Glucose uptake Glo Assay (Promega #J1342) as per the manufacturer’s instructions. Bioluminescence was measured in duplicate using the Biotek Synergy H1 plate reader.

### 4.11. Scratch Wound Migration Assay

ARPE-19 cells were seeded in 6-well plates to confluence and serum starved for 2 days before mechanical scratching with a p200 pipette tip. Cells were gently washed with PBS before adding TGFβ2 diluted in serum-free culture medium. Phase-contrast images were taken at 0, 24 and 48 h post-scratch using the EVOS M5000 Cell Imaging System (Life Technologies, ThermoFisher Scientific). The area of wound closure was quantified using FIJI [62] for the same scratched region in each well and quantified as a percentage change in wound closure area.

### 4.12. Statistical Analysis

All statistical analyses were performed using GraphPad Prism 8.3.0. Comparisons of parametric data were analyzed by one-way ANOVA with Tukey’s post-hoc analysis or two-way ANOVA with post hoc analysis using the Sidak method. The Brown–Forsythe and Bartlett’s test were used to assess variance homogeneity. For data with unequal variances, the Dunnett’s T3 multiple comparisons post-hoc test was applied. Mitochondrial network parameters were analyzed using unpaired Student’s *t*-test. Statistical significance was considered when *p* < 0.05 and all data are displayed as mean ± SEM. * *p* < 0.05, ** *p* < 0.01, *** *p* < 0.001 and **** *p* < 0.0001.

## Figures and Tables

**Figure 1 ijms-22-04701-f001:**
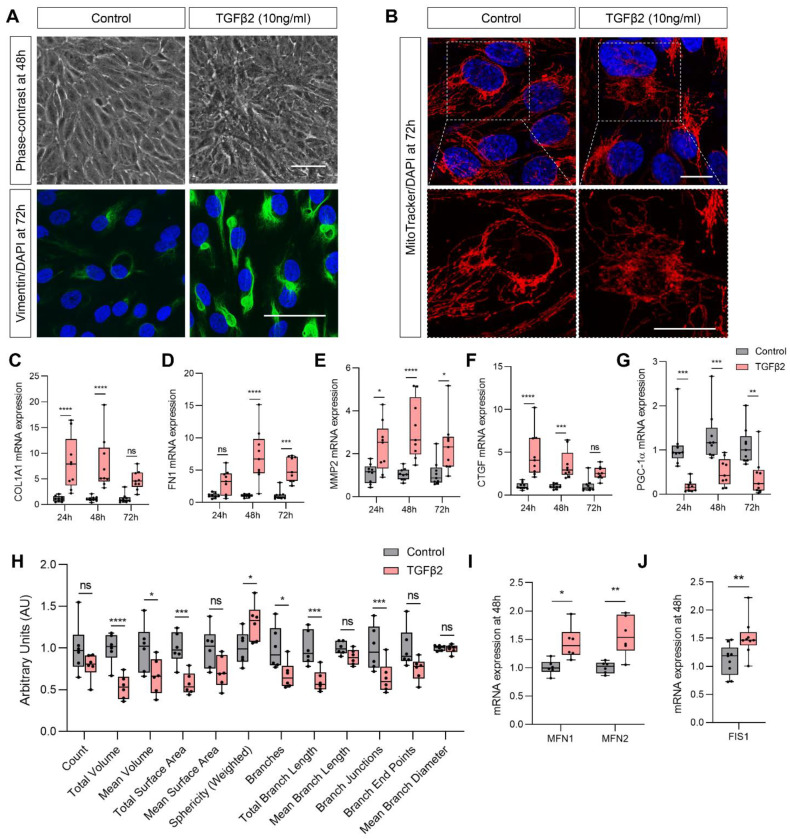
TGFβ2-induced epithelial-mesenchymal transition (EMT) is accompanied by mitochondrial dysfunction. ARPE-19 cells were treated with TGFβ2 for up to 72 h. (**A**) Representative phase-contrast images depicting cellular morphology at 48 h (*n* = 3, scale bar is 50 μm) and representative immunofluorescence images of the mesenchymal marker, vimentin (green) and DAPI (nuclei, blue) at 72 h (*n* = 3, scale bar is 50 μm). (**B**) Representative images showing mitochondrial morphology using confocal microscopy imaging of MitoTracker Orange (red) and DAPI (nuclei, blue) at 72 h (*n* = 3, scale bar is 20 μm) for control and TGFβ2-treated cells and corresponding magnified views of selected areas (scale bar is 20 μm). Quantification of mesenchymal genes (**C**) *COL1A1*, (**D**) *FN1*, (**E**) *MMP2*, (**F**) *CTGF* and (**G**) *PGC-1α* by qPCR at 24, 48 and 72 h with or without TGFβ2 treatment (*n* = 9, two-way ANOVA with Sidak’s post-hoc analysis). (**H**) Confocal images were analyzed by automated processing to quantify mitochondrial shape and network parameters (*n* = 6, unpaired *t*-test). Quantification of mitochondrial fusion gene expression following treatment with or without TGFβ2 at 48 h for fusion genes (**I**) *MFN1* and *MFN2* (*n* = 6, two-way ANOVA with Sidak’s post-hoc analysis) and the mitochondrial fission gene (**J**) *FIS1* (*n* = 9, unpaired *t*-test). Error bars are means ± SEM. * *p* ≤ 0.05; ** *p* ≤ 0.01; *** *p* ≤ 0.001; **** *p* ≤ 0.0001; ns, not significant.

**Figure 2 ijms-22-04701-f002:**
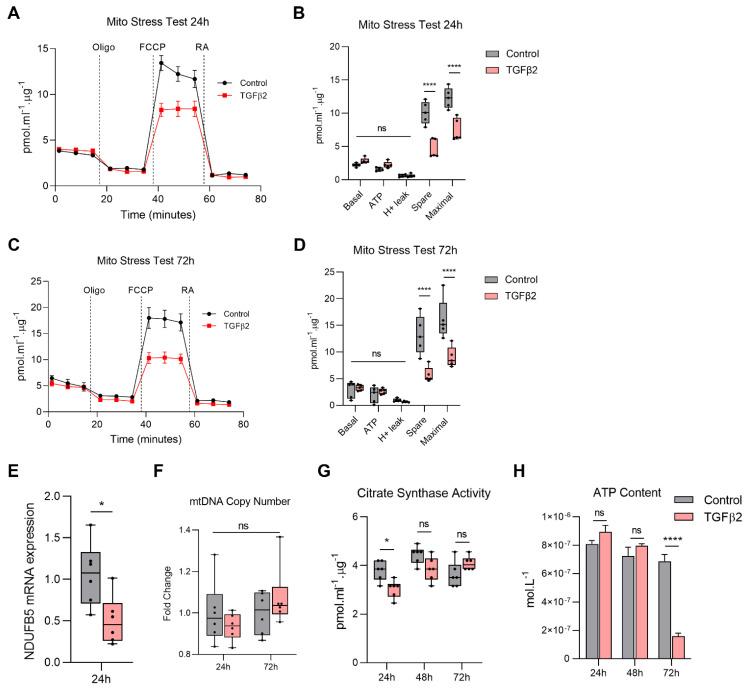
TGFβ2 reduces mitochondrial respiration in RPE. Real-time measurement of oxygen consumption rate (OCR) using the Seahorse XF24 BioAnalyzer to assess OXPHOS parameters: basal respiration, ATP-linked respiration, proton leak, spare respiratory capacity and maximal respiration based on responses to drug injections of oligomycin (Oligo), FCCP and rotenone and antimycin A (RA) at (**A**,**B**) 24 h and (**C**,**D**) 72 h in ARPE-19 treated with and without TGFβ2 (*n* = 5, one-way ANOVA with Tukey’s post-hoc analysis). (**E**) Quantification of *NDUFB5* gene expression following treatment with or without TGFβ2 at 24 h (*n* = 6, unpaired *t*-test). (**F**) Quantification of mitochondrial DNA (mtDNA) copy number using qPCR with and without TGFβ2 for 24 and 72 h (*n* = 6, one-way ANOVA with Tukey’s post-hoc analysis). Error bars are means ± SEM. * *p* ≤ 0.05; ns, not significant. (**G**) Quantitation of citrate synthase activity (*n* = 6) and (**H**) Intracellular ATP content at 24, 48 and 72 h with or without TGFβ2 (*n* = 3, one-way ANOVA with Tukey’s post-hoc analysis). Error bars are means ± SEM. * *p* ≤ 0.05; **** *p* ≤ 0.0001; ns, not significant.

**Figure 3 ijms-22-04701-f003:**
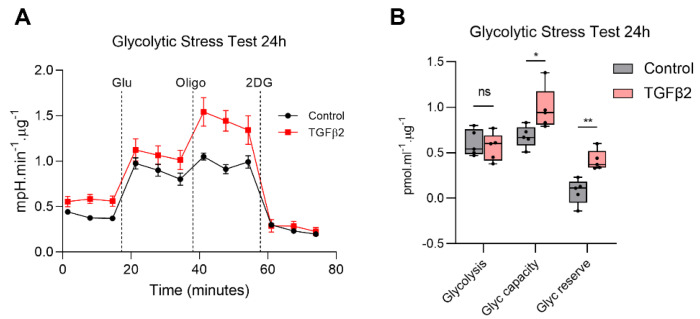
TGFβ2 enhances glycolytic reserve in RPE and increases glycolysis gene expression. Real-time measurement of extracellular acidification rate (ECAR) using the Seahorse XF24 BioAnalyzer to assess glycolytic function parameters: glycolysis, glycolytic capacity and glycolytic reserve based on responses to drug injections of glucose (Glu), oligomycin (Oligo) and 2-deoxyglucose (2DG) at (**A**,**B**) 24 h (**C**,**D**) 72 h treated with and without TGFβ2 (*n* = 5, one-way ANOVA with Tukey’s post-hoc analysis). (**E**) Glucose uptake at 24 h with or without TGFβ2 (*n* = 6, unpaired *t*-test) in relative light units (RLU). qPCR analysis of (**F**) glucose transporters at 24 h, (**G**) lactate transporters at 24 h and glycolysis gene expression (**H**) *PFKFB3*, (**I**) *PKM2* and (**J**) *LDHA* at 24, 48 and 72 h with or without TGFβ2 (*n* = 9 for *PFKFB3* otherwise *n* = 6, one-way ANOVA with Tukey’s post-hoc analysis). Error bars are means ± SEM. * *p* ≤ 0.05; ** *p* ≤ 0.01; *** *p* ≤ 0.001; ns, not significant.

**Figure 4 ijms-22-04701-f004:**
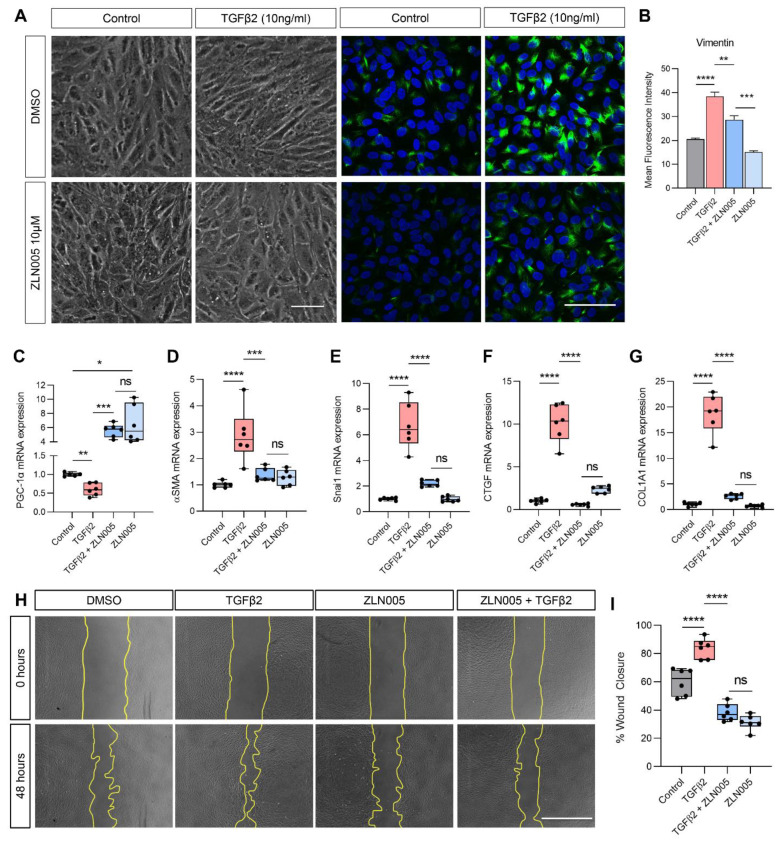
TGFβ2 suppresses PGC-1α and ZLN005, the selective small molecular activator of PGC-1α, blocks TGFβ2-induced EMT. (**A**) Representative phase-contrast microscopy and immunofluorescence for vimentin (green) with DAPI at 72 h with or without TGFβ2 and/or ZLN005 (*n* = 3, scale bar is 50 μm) and (**B**) quantification of mean fluorescence intensity. Quantification of (**C**) *PGC-1α* and mesenchymal genes (**D**) *α-SMA*, (**E**) *Snai1*, (**F**) *CTGF* and (**G**) *COL1A1* by qPCR at 24 h with or without TGFβ2 and/or ZLN005 treatment (*n* = 6). (**H**) Representative phase-contrast microscopy images for the scratch wound migration assay at 48 h with or without TGFβ2 and/or ZLN005 (*n* = 6, scale bar is 100 μm) and (**I**) quantification of percentage wound closure (*n* = 6). Significance was determined by one-way ANOVA with Tukey’s or Dunnet’s T3 post-hoc analysis. Error bars are means ± SEM. * *p* ≤ 0.05; ** *p* ≤ 0.01; *** *p* ≤ 0.001; **** *p* ≤ 0.0001; ns, not significant.

**Figure 5 ijms-22-04701-f005:**
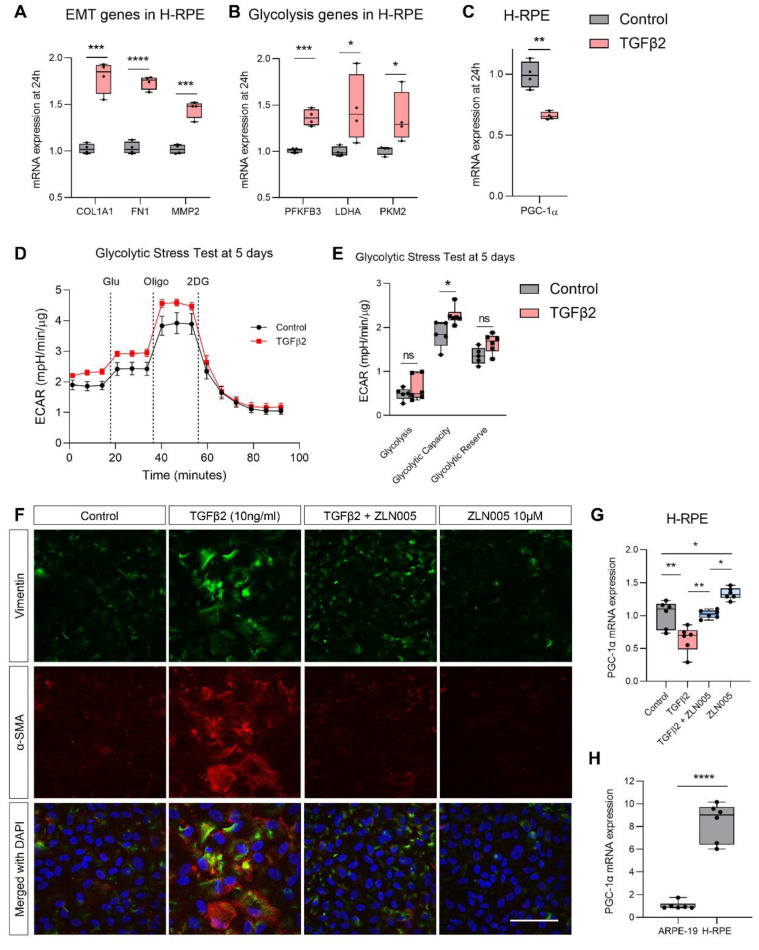
TGFβ-induced EMT in primary human RPE is blocked by ZLN005. Quantification of mesenchymal genes (**A**) *COL1A1*, *FN1*, *MMP2*, (**B**) glycolytic genes (*PFKFB3*, *LDHA*, *PKM2*) and (**C**) *PGC-1α* by qPCR at 24 h with or without TGFβ2 treatment (*n* = 4, unpaired *t*-test). Real-time measurement of extracellular acidification rate (ECAR) using the Seahorse XFe96 BioAnalyzer to assess glycolytic function parameters: glycolysis, glycolytic capacity and glycolytic reserve based on responses to drug injections of glucose (Glu), oligomycin (Oligo) and 2-deoxyglucose (2DG) at (**D**,**E**) 24 h treated with and without TGFβ2 (*n* = 5, unpaired *t*-test). (**F**) Representative immunofluorescence for vimentin (green) and α-SMA (red) with DAPI at 72 h with or without TGFβ2 and/or ZLN005 (*n* = 3, scale bar is 50 μm). Quantification of (**G**) *PGC-1α* in H-RPE by qPCR at 24 h with or without TGFβ2 and/or ZLN005 treatment and (**H**) comparison of *PGC-1α* levels between control ARPE-19 and H-RPE (*n* = 6). Error bars are means ± SEM. * *p* ≤ 0.05; ** *p* ≤ 0.01; *** *p* ≤ 0.001; **** *p* ≤ 0.0001; ns, not significant.

**Table 1 ijms-22-04701-t001:** Primer sequences for qPCR.

Gene Symbol	Gene Name	Forward Sequence (5′-3′)	Reverse Sequence (5′-3′)
*A-SMA*	Alpha-smooth muscle actin	AAAAGACAGCTACGTGGGTGA	GCCATGTTCTATCGGGTACTTC
*B2M*	Beta-2 microglobulin	TTCTGGTGCTTGTCTCACTGA	CAGTATGTTCGGCTTCCCATTC
*COL1A1*	Collagen, type I, alpha 1	GTGCGATGACGTGATCTGTGA	CGGTGGTTTCTTGGTCGGT
*CTGF*	Connective tissue growth factor	CAGCATGGACGTTCGTCTG	AACCACGGTTTGGTCCTTGG
*FIS1*	Fission 1	TGACATCCGTAAAGGCATCG	CTTCTCGTATTCCTTGAGCCG
*FN1*	Fibronectin 1	CGGTGGCTGTCAGTCAAAG	AAACCTCGGCTTCCTCCATAA
*GLUT1*	Glucose Transporter 1 (SLC2A1)	GGCCAAGAGTGTGCTAAAGAA	ACAGCGTTGATGCCAGACAG
*GLUT12*	Glucose Transporter 12 (SLC2A12)	GAGGCTGCGGCATGTTTAC	CCAAGTTCATAACCCACCAGG
*GLUT3*	Glucose Transporter 3 (SLC2A3)	GCTGGGCATCGTTGTTGGA	GCACTTTGTAGGATAGCAGGAAG
*LDHA*	Lactate dehydrogenase A	GGCCTGTGCCATCAGTATCT	GGAGATCCATCATCTCTCCC
*MCT1*	Monocarboxylic acid transporter 1 (SLC16A1)	AGGTCCAGTTGGATACACCCC	GCATAAGAGAAGCCGATGGAAAT
*MCT3*	Monocarboxylate transporter, member 3 (SLC16A3)	CCATGCTCTACGGGACAGG	GCTTGCTGAAGTAGCGGTT
*MFN1*	Mitofusin 1	TGCCCTTCACATGGACAAAG	CTCTGTAGTGACATCTGTGCC
*MFN2*	Mitofusin 2	ATGTGGCCCAACTCTAAGTG	CACAAACACATCAGCATCCAG
*MMP2*	Matrix metalloproteinase-2	CTTCCAAGTCTGGAGCGATGT	TACCGTCAAAGGGGTATCCAT
*NDUFB5*	NADH dehydrogenase [ubiquinone] 1 beta subcomplex, 5, 16 kDa	CACTCGCCTCGGATTTGG	CGCCTGTCATAGAATCTAGAAGG
*PFKFB3*	6-phosphofructo-2-kinase/fructose-2,6-biphosphatase 3	CAGTTGTGCCTCCAATATC	GGCTTCATAGCAACTGATCC
*PGC-1A*	Peroxisome proliferator-activated receptor gamma, coactivator 1 alpha (PPARGC1A)	GTCACCACCCAAATCCTTAT	ATCTACTGCCTGGAGACCTT
*PKM2*	Pyruvate kinase muscle isozyme M2	CAAAGGACCTCAGCAGCCATGTC	GGGAAGCTGGGCCAATGGTACAGA
*PPIH*	Peptidylprolyl isomerase H (cyclophilin H)	CCCCAACAATAAGCCCAAG	CACCACCAAGAAGAAGGGAA
*SNAI1*	Snail 1	TCGGAAGCCTAACTACAGCGA	AGATGAGCATTGGCAGCGAG
*TBP*	TATA-binding protein	TGCACAGGAGCCAAGAGTGAA	CACATCACAGCTCCCCACCA

## Data Availability

The data presented in this study are available on request from corresponding author.

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
