# Peer review of "Suppression of PGC-1α Drives Metabolic Dysfunction in TGFβ2-Induced EMT of Retinal Pigment Epithelial Cells"

_ijms, 2021, doi:10.3390/ijms22094701_

Round 1
Reviewer 1 Report
The paper is well written, very clear and impactful. I believe it may be absolutely acceptable for publication
Author Response
We thank the reviewer for the positive review and kind words.
Reviewer 2 Report
The authors presented an interesting work on metabolic dysfunction in TGFβ2-induced EMT of retinal pigment epithelial (RPE) cells.The authors highlight that TGFβ2 alters the structure and the respiratory capacity of mitochondria, thereby reducing the production of ATP.
All these observations have been related to the reduced expression of PGC1α in cells treated with TGFβ2. This last conclusion was supported, albeit indirectly, by the attenuation of the TGFβ2 action and, correspondingly, of its effects, upon treatment of cells with ZLN005, a selective small molecule activator of PGC-1α. Experimental data were obtained using the ARPE-19 cells and primary RPE cells.
In general, the data are consistent with the authors conclusions, and the results are almost rigorous enough to allow the strong conclusions made.
An important concern is on the role of PGC1α in the reduction of EMT in TGFβ2-treated cells. The authors refer to previous articles in which it is asserted that ZLN005, the selective small molecular activator of PGC-1α, effectively increased PGC-1α gene expression and enhanced mitochondrial respiration. While the authors showed that treatment of cells with TGFβ2 drastically reduced PGC-1α expression, the effect of ZLN005 on expression and transcriptional activity PGC-1α in EMT-induced RPE cells was not shown. This experimental data is fundamental to support the evidence of PGC-1α's role in reversing the TGFβ2 effects.
The discussion on the role of ZLN005 / PGC1a should be implemented with other assumptions. ZLN005 could induce the expression of Sirtuin 1 (Li et al., Exp Cell Res 2016; 345:25-36). Moreover, SIRT1 could negatively affect the TGFβ2/SMAD transduction pathway (Simic et al, Cell Rep. 2013 3(4); Bugyei-Twum et al, Cardiovasc Res. 2018; 114:1629–1641).
Author Response
We thank the reviewer for their interest in our work and valuable insights. We agree with the reviewer’s comments. As suggested, we have now added data showing that ZLN005 upregulates PGC-1α gene expression in both ARPE-19 (6-fold increase; Figure 4C) and H-RPE cells (1.3-fold increase; Figure 5G). Importantly, we show that the elevated PGC-1α levels induced by ZLN005 are maintained even in the presence of TGFβ2. We also showed that the basal levels of PGC-1α in control H-RPE is significantly higher than in ARPE-19 (8-fold higher in H-RPE; Figure 5H) which may help to explain why there is more robust upregulation of PGC-1α levels by ZLN005 in ARPE-19 compared to H-RPE.
We thank the reviewer for enhancing our discussion on the mechanistic role of ZLN005 in blocking EMT. We have now added these references in the discussion section.
Round 2
Reviewer 2 Report
The authors answered all the points raised in the first review. The manuscript is now suitable for publication in IJMS.